# Meta-Analysis of Hypoxic Transcriptomes from Public Databases

**DOI:** 10.3390/biomedicines8010010

**Published:** 2020-01-09

**Authors:** Hidemasa Bono, Kiichi Hirota

**Affiliations:** 1Database Center for Life Science (DBCLS), Joint Support-Center for Data Science Research, Research Organization of Information and Systems, 1111 Yata, Mishima, Shizuoka 411-8540, Japan; 2Department of Human Stress Response Science, Institute of Biomedical Science, Kansai Medical University, Hirakata 573-1010, Japan

**Keywords:** hypoxia, transcriptome, RNA-seq, ChIP-seq, public database, meta-analysis

## Abstract

Hypoxia is the insufficiency of oxygen in the cell, and hypoxia-inducible factors (HIFs) are central regulators of oxygen homeostasis. In order to obtain functional insights into the hypoxic response in a data-driven way, we attempted a meta-analysis of the RNA-seq data from the hypoxic transcriptomes archived in public databases. In view of methodological variability of archived data in the databases, we first manually curated RNA-seq data from appropriate pairs of transcriptomes before and after hypoxic stress. These included 128 human and 52 murine transcriptome pairs. We classified the results of experiments for each gene into three categories: upregulated, downregulated, and unchanged. Hypoxic transcriptomes were then compared between humans and mice to identify common hypoxia-responsive genes. In addition, meta-analyzed hypoxic transcriptome data were integrated with public ChIP-seq data on the known human HIFs, HIF-1 and HIF-2, to provide insights into hypoxia-responsive pathways involving direct transcription factor binding. This study provides a useful resource for hypoxia research. It also demonstrates the potential of a meta-analysis approach to public gene expression databases for selecting candidate genes from gene expression profiles generated under various experimental conditions.

## 1. Introduction

The development of high-throughput sequencing technology has enabled cost-effective reading of tens of millions of base pairs in a single run. RNA-seq takes advantage of this technology to elucidate the expression profiles of genes and transcriptomes assayed under particular conditions by producing counts of sequences corresponding to genes of interest.

Published transcriptome data have been archived to two large public databases (DBs), the Gene Expression Omnibus (GEO) in the US National Center for Biotechnology Information (NCBI) [1] and ArrayExpress (AE) in the European Bioinformatics Institute (EBI) [2]. The number of records in these DBs is now over two million in samples and near a hundred thousand in data series. They are freely accessible and thus ready to be reused for data-driven research. Nevertheless, large-scale comparison among archived data has not often been carried out because of the technical challenges presented by the magnitude, complexity, and cumbersome nature of the data. Even if very large amounts of data can be successfully downloaded, it can be difficult to interpret data from different laboratories, as experimental protocols used are not uniform.

Even though oxygen is an essential molecule for life support, higher organisms such as mammals do not have a mechanism to biosynthesize oxygen in the body. Organs and tissues are regularly exposed to risk of “oxygen deficiency”, and living organisms have evolved mechanisms to respond to hypoxia. The supply and consumption of oxygen determine its intracellular partial pressure, which is kept in a relatively narrow range. Oxygen deficiency (hypoxia) and oxygen excess (hyperoxia) cause an adaptive response to maintain oxygen homeostasis at the cellular level. This includes adjustments to the energy metabolism system according to changes in oxygen partial pressure. In addition, higher organisms that are anatomically complex have specialized mechanisms to acquire necessary and sufficient oxygen for all cells. Proper functioning and regulation of these systems require the coordinated expression of many genes, and this is thought to be controlled by transcription factors known as hypoxia-inducible factors (HIFs, HIF-1 and HIF-2) [3]. The differential regulation of gene expression by HIF-1 and HIF-2 has been studied. Both HIF-1 and HIF-2 control the gene expression of glucose transporter 1 (GLUT1) and vascular endothelial growth factor A (VEGFA). Gene expression of glycolytic enzymes such as hexokinases (HK1 and HK2), phosphofructokinase (PFK), fructose-bisphosphate aldolase A (ALDOA), phosphoglycerate kinase 1 (PGK1), and lactate dehydrogenase (LDHA) controlled by mainly HIF-1. On the other hand, gene expression of erythropoietin (EPO) and POU5F1 (OCT4) is HIF-2-dependent [4].

Various groups have published comprehensive data on gene expression in response to hypoxic stress, but have not produced uniform results, and because of this the pathways involved are not entirely clear. In addition, there may be other data on hypoxia-related genes that have not been reported as such, due to the main experimental target not being hypoxic stress [5]. Collective analysis of gene expression and transcription factor binding information obtained from multiple studies that also introduces comparative analysis between species has not previously been attempted in this field. Further, although the analysis of many comprehensive datasets should enable the elucidation of novel control pathways, comprehensive methods to achieve this are lacking.

Based on the above background, this study aimed to develop a comparative data analysis method to identify expression fluctuations and transcription factor binding regions associated with hypoxic stress, and to use it to perform a meta-analysis of available data in the public DBs, and gene functional analyses. We first curated hypoxic RNA-seq transcriptome data from GEO and AE to make a list of hypoxia-normoxia transcriptome pairs. We then did systematic transcriptome quantification analysis on the datasets we collected. By introducing a new metric that counts instances of upregulation, downregulation, or unresponsiveness of genes, we determined the average effect of hypoxia on genes under different experimental conditions. We also compared these values between genes orthologous in humans and mice. In addition, we examined data from chromatin immunoprecipitation sequencing (ChIP-seq), which comprehensively elucidates the regions in which transcription factors bind to genomic DNA.

## 2. Materials and Methods

### 2.1. Curation of Public Gene Expression Data

In order to screen hypoxia-related gene expression data from public databases, we used a graphical web tool called All Of gene Expression (AOE; https://aoe.dbcls.jp/) [6]. AOE not only integrates metadata from the NCBI Gene Expression Omnibus (GEO) [1], EBI ArrayExpress (AE) [2], and DDBJ Genomic Expression Archive (GEA) [7], but also those of RNA-seq data archived only in the Sequence Read Archive (SRA) [8]. Conventional search by the keywords ‘hypoxia’ or ‘hypoxic’ in AOE was adopted to scan the databases initially. The curation of the data, which included paired ‘hypoxia’ and ‘normoxia’ experiment entries, and descriptions of cell line and experimental conditions (oxygen concentration) used, was done manually.

In our previous pilot study, we investigated the human and mouse data produced by Affymetrix GeneChip [9], but the subsequent accumulation of RNA-seq data enabled this study to be focused only on RNA-seq. In order to minimize the noise from different sequencing platforms, it was necessary to exclude data from older sequencing platforms. Consequently, nearly all data were the product of Illumina sequencing platforms. The complete lists of these paired data are available from figshare (human: https://doi.org/10.6084/m9.figshare.5811987.v2; mouse: https://doi.org/10.6084/m9.figshare.9948158.v1).

### 2.2. Gene Expression Quantification

After we searched hypoxia-related entries using AOE, corresponding run data were downloaded from SRA in the DDBJ FTP site (ftp://ftp.ddbj.nig.ac.jp/). Since the downloaded data were in the SRA format, these files were transferred to FASTQ formatted files for expression quantification using the fasterq-dump program in the SRA Toolkit (https://www.ncbi.nlm.nih.gov/sra/docs/toolkitsoft/). Both single-end and paired-end reads were re-used for the analysis. RNA-seq reads were then quantified using ikra (v1.2.0) [10], an RNA-seq pipeline centered on Salmon [11]. Ikra automates the RNA-seq data analysis process, which includes quality control of reads (Trim Galore version 0.4.1 [12] with Cutadapt version 1.9.1 [13]) and transcript quantification (Salmon version 0.14.0 with reference transcript sets in GENCODE release 30 for human and M21 for mouse). These tools were used with default parameters in ikra. The workflow presented here was selected because we aimed to extract hypoxia inducible genes from heterogeneous RNA-seq data (single-end and paired-end) archived in SRA from various laboratories by counting upregulated and downregulated genes. In this study, the data acquisition and quality control process took around six weeks for the current data set. Processed transcript quantification from RNA-seq data was also uploaded to figshare and are publicly available (human: https://doi.org/10.6084/m9.figshare.9948170.v1; mouse: https://doi.org/10.6084/m9.figshare.9948200.v1).

Where hypoxia and normoxia transcriptome data were paired, the ratio of all gene pairs (termed the HN ratio) was calculated.
(1)$$\mathrm{HN}\;\mathrm{ratio}=\frac{\left(\mathrm{Gene}\;\mathrm{expresion}\;\mathrm{value}\;\mathrm{in}\;\mathrm{hypoxia}\right)\;+\;1}{\left(\mathrm{Gene}\;\mathrm{expression}\;\mathrm{value}\;\mathrm{in}\;\mathrm{normoxia}\right)\;+\;1}$$

Values of HN ratios for all paired samples were then classified into three groups. When the HN ratio was over the threshold for upregulation, the gene was regarded as upregulated. Similarly, when the HN ratio was below the threshold for downregulation, the gene was regarded as downregulated. If the gene was labeled as neither upregulated nor downregulated, it was classified as ‘unchanged’. Finally, the numbers of counts for up, down, and unchanged were calculated for all genes. For the up-/downregulated gene classification, several thresholds were tested to optimize the calibration. For this study, we adopted a two-fold threshold after several parameters (1.5, 2, 5, and 10-fold) was tested to classify up/downregulated genes. The number of paired samples in which genes were up/down regulated was counted for all genes in the human genome.

For the evaluation of hypoxia-inducible genes, a hypoxia-and-normoxia score (HN-score) was calculated for all genes in humans and mice respectively. HN-score was the count of (count of human RNA-seq UP] − (count of human RNA-seq DOWN]. HN-scores for all genes were also calculated in mice. Orthologous genes between humans and mice and the functional annotations of genes were downloaded from Ensembl Biomart [14]. Full lists of counts (up/down/unchanged) with HN-scores for all genes were accessible from figshare (human: https://doi.org/10.6084/m9.figshare.5812710.v3; mouse: https://doi.org/10.6084/m9.figshare.9948233.v2).

All codes used for processing the data are freely available from GitHub (https://github.com/bonohu/chypoxia/).

### 2.3. Meta-Analysis of ChIP-Seq Data

Public ChIP-seq data were collected, curated, and pre-calculated for reuse in the ChIP-Atlas database [15]. Average MACS2 scores for all genes were retrieved using the ‘Target Genes’ tool in ChIP-Atlas for hypoxia-inducible factor 1-α (HIF1A) and Endothelial PAS domain-containing protein 1 (EPAS1, also known as hypoxia-inducible factor-2α (HIF-2α)) as ‘Antigens’ with ± 5k for the ‘Distance from TSS’ parameter. For the integration of ChIP-seq data into RNA-seq data produced above, the names of genes were used to join two datasets.

### 2.4. Visualization and the Integrated Functional Analysis of Genes

For producing scatter plots, we used TIBCO Spotfire Desktop version 7.6.0 (TIBCO Spotfire, Inc., Palo Alto, CA, USA) with TIBCO Spotfire’s “Better World” program license (http://spotfire.tibco.com/better-world-donation-program/) in this study.

Metascape was used for the gene set enrichment analysis [16]. Conventional ‘express analysis’ in Metascape was used to draw histograms. In the gene set enrichment analysis of genes upregulated and downregulated in hypoxic transcriptomes (Appendix A), the HN-scores described above were used for the extraction of a gene list for Metascape input. Two lists of genes were generated by extracting genes whose HN-score was over (or below) a threshold, where roughly 1% of all genes could be listed. These HN-score thresholds were 32 for upregulation (374 genes) and −34 for downregulation (324 genes), respectively.

## 3. Results

### 3.1. Curation of Hypoxic Transriptome Data in Public Databases

We initially mined hypoxia-related gene expression data from public databases using an integrated graphical web tool for gene expression data called AOE, which has been maintained as an index of public gene expression databases. Using the conventional keyword search by ‘hypoxia’ in AOE, we showed that the number of paired samples was very few in most model organisms except humans and mice.

Pairs of samples before and after hypoxic stress were made after careful curation of dataset descriptions. We were able to obtain 128 pairs from 35 data series in humans and 53 pairs from 10 data series in mice in 2018. The complete list of pairs in RNA-seq data by Illumina sequencers is also accessible from figshare (human: https://doi.org/10.6084/m9.figshare.5811987.v2; mouse: https://doi.org/10.6084/m9.figshare.9948158.v1).

The overall procedure of the work described above is depicted in Figure 1.

### 3.2. Meta-Analysis of Hypoxia-Responsive Genes

After the quantification of gene expression from RNA-seq data, the number of conditions under which each gene was upregulated, downregulated, and unchanged were counted. The reason for this three-way categorization is that the data are highly series-specific owing to various cell lines and experimental conditions. Complete lists of the meta-analyzed results are accessible from figshare (human: https://doi.org/10.6084/m9.figshare.5812710.v3; mouse: https://doi.org/10.6084/m9.figshare.9948233.v2).

In order to visualize differentially expressed genes, we introduced a value called HN-score. HN-score is the number of UP counts minus the number of DOWN counts and was calculated for all genes. Using this score, we were able to quantify the degree to which each gene was affected by hypoxia. For example, *VEGFA* had 92 UP, 6 DOWN, and 30 unchanged counts. Its HN-score was thus 86 (= 92 − 6).

Following this, genes orthologous between humans and mice were related utilizing an orthologous gene table generated from Ensembl Biomart. This operation yields a list of genes concurrently upregulated in humans and mice. In order to sort the table, the sum of human and mouse HN-scores was calculated (called total HN-score). For example, the score for *VEGFA* (human) was 86 and that for *Vegfa* (mouse) was 23, so the total HN-score for *VEGF* gene was 109. Table 1 shows the top 25 genes with high total HN-score, and a complete merged list is available from figshare (https://doi.org/10.6084/m9.figshare.9958169.v1).

Figure 2 visualizes this table as a scatter plot of HN-score values for humans and mice. Genes located in the upper right are those upregulated after hypoxia both in humans and mice. Genes in that category included previously reported typical hypoxia-responsive genes, for example *PGK1*, *VEGFA*, and *EGLN3*, supporting the validity of our method. On the other hand, genes downregulated both in humans and mice included some not previously identified as hypoxia-related.

We then performed set enrichment analysis using Metascape. The analysis clearly revealed that some genes upregulated in many samples with high HN-scores are well-known hypoxia-responsive genes, while some are not (Appendix A). The latter can be novel candidates for hypoxia-responsive genes to study signaling pathways in hypoxia research. Metascape also clearly depicted the functions of downregulated genes with low HN-scores (Appendix A). Those genes were apparently related to DNA repair and replication [17].

### 3.3. Integration of Meta-Analyzed ChIP-Seq Data

In order to investigate expression regulation of genes of interest by direct transcription factor binding, we studied ChIP-seq data. In addition to the public nucleotide sequence databases, ChIP-seq data processed by the MACS2 program are also available from the ChIP-Atlas database, which is a database for meta-analysis results from publicly available ChIP-seq data [15]. Thus, we retrieved ChIP-seq data for hypoxia-inducible factor 1-alpha (HIF1A) and endothelial PAS domain-containing protein 1 (EPAS1, also known as hypoxia-inducible factor-2alpha (HIF-2alpha)) from ChIP-Atlas.

We integrated meta-analysis results of human RNA-seq (hypoxic transcriptome) described above and human ChIP-seq data for HIF1A and EPAS1. Then, we visualized the results by conventional scatterplot (Figure 3). The source data are available from figshare (https://doi.org/10.6084/m9.figshare.9958181.v2). In addition to reported hypoxia-responsive genes such as *ANKRD37* [18], novel HIF1-target candidate genes were found with high HN-score and high ChIP-seq score (Figure 3A). In Figure 3A, genes with high HN-score but low ChIP-seq score were also found, and these may belong to the non-HIF1 regulation pathway. *NDRG1*, which encodes a cytoplasmic protein involved in stress responses, hormone responses, cell growth, and differentiation, is a typical gene with such a pattern. Furthermore, we found HIF1-specific and HIF2-specific genes in the scatter plots for HIF1A (Figure 3A) and EPAS1 (Figure 3B). Genes with high ChIP-scores in EPAS1 but not in HIF1A could be HIF2 targets, but there were no distinct genes in this category.

In order to study the differences between HIF1A and EPAS1 more thoroughly, we then investigated genes with high MACS2 score for HIF1A and EPAS1 (with positive HN-score). Surprisingly, the top 100 genes for HIF1A and EPAS1 were exactly the same, and we therefore increased the number of genes to 300. The top 300 genes for HIF1A (https://doi.org/10.6084/m9.figshare.9958235.v1) and EPAS1 (https://doi.org/10.6084/m9.figshare.9958250.v1) were generated and compared to identify the differences. The two gene lists were analyzed and visualized using the ‘calculate and draw custom Venn diagrams’ website (http://bioinformatics.psb.ugent.be/webtools/Venn/; Figure 4A). Gene set enrichment analysis of the intersection of the two gene lists revealed typical features of hypoxia-inducible genes (Figure 4B), while analysis of the HIF1A-specific and EPAS1-specific portions also showed interesting features (Figure 4C,D). Genes with the functional annotations ‘M00001: glycolysis (Embden-Meyerhof pathway)’ and ‘GO:0031167: rRNA methylation’ were enriched among HIF1A-specific genes (Figure 4C). This preferential regulation of glycolysis by HIF1A) was previously described from microarray data [19]. On the other hand, genes with ‘GO:0040008: regulation of growth’ and ‘GO:0003151: outflow tract morphogenesis’ were enriched among EPAS1-specific genes (Figure 4D). These observations may reflect target gene differences between these two transcription factors.

## 4. Discussion

While over two million transcriptome samples have been archived in public databases (NCBI Gene Expression Omnibus and EBI ArrayExpress), these data were reported by various laboratories and are thus derived from different populations, analysis platforms, and sampling conditions. The resultant variability can compromise meaningful comparisons, and it is indispensable to curate the data manually to do meta-analysis for the study of specific biological stresses.

Using AOE, we made a complete survey of hypoxia-related gene expression data in the public databases. The large quantity of relevant data available made it possible to do meta-analysis at the level of transcriptome sequences. In our previous study, only 23 hypoxia-normoxia pairs could be analyzed due to a lack of available human RNA-seq, and no relevant murine RNA-seq data could be found. However, although the amount of microarray data available is much greater, most RNA-seq data are from Illumina platforms (for example HiSeq2500, HiSeq2000, and NextSeq500), simplifying comparison. Thus, we decided to use data only from RNA-seq for our meta-analysis on the hypoxic transcriptome in the public databases.

We manually curated data by reference to recorded metadata and made pairs of hypoxia-normoxia data from human and mouse cell lines. It is often troublesome to handle ratio data that contains a very large number of columns. Thus, we tentatively set the threshold for upregulation and downregulation in hypoxia, and reduced the ratio information for all samples into a cumulative upregulation/downregulation count. This conversion made it substantially easier to interpret genes biologically. After optimization, we determined a threshold of two-fold for this study to filter genes for up/downregulated although it is not a process to extract statistically significant differentially expressed genes.

Gene set enrichment analysis for genes with high HN-scores showed that genes involved in ‘HIF-1 alpha transcription factor network’ (GSEA Gene Set: PID_HIF1_TFPATHWAY) and ‘response to oxygen levels’ (Gene Ontology: GO:0070482) were reasonably enriched (Appendix A). This evidence justifies our meta-analysis on hypoxic transcriptomes. Detailed analyses of hypoxia-inducible genes in humans and mice (Table 1) identified a series of hypoxia-inducible genes in a data-driven manner.

In addition, our integration of ChIP-seq into RNA-seq data added substantial information on hypoxia-inducible genes (Figure 3). HIF binding directly mediates gene upregulation, but not for gene downregulation as described in previous works [20,21]. High ChIP-seq scores, based on the MACS2 program via the ChIP-Atlas database, indicated genes whose regulatory regions were directly bound by transcription factors. For example, *ANKRD37* showed both high HN-score and high ChIP-seq score (Figure 3A), while *NDRG1* had a high HN-score but a low ChIP-seq score. By differentiating genes in this way, we can generate additional hypotheses about pathways regulating differential regulation until the hypothesis that the transcription factor binds near to the transcription start site (TSS) is valid. For example, EPAS1/HIF2A preferentially binds to distant regulatory regions [22]. In this case, we cannot use meta-analyzed ChIP-seq data for further analyses. In other cases where transcription factor binds to genomic region near TSS, the integration of meta-analyzed ChIP-seq and RNA-seq data clearly filtered genes with direct and indirect regulation, and this information can be a valuable resource for the functional analysis of hypoxia-inducible genes.

Our future work will involve detailed analyses of co-upregulated genes across many more species, which will be possible after the compilation of such transcriptome data. In view of the insights enabled by the integration of ChIP-seq with RNA-seq data, another potential future approach is the inclusion of additional types of omics data.

All data described in the manuscript are archived in figshare as a collection (Bono, H. figshare https://doi.org/10.6084/m9.figshare.c.4690397.v1 (2019)). Source codes to replicate our study are freely available from GitHub (https://github.com/bonohu/chypoxia).

## Figures and Tables

**Figure 1 biomedicines-08-00010-f001:**
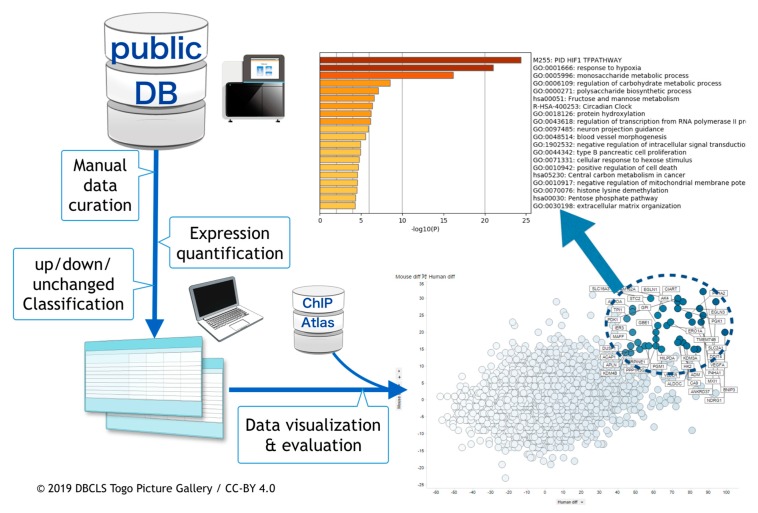
Schematic view of hypoxic transcriptome meta-analysis. Public databases were searched and hypoxia-related RNA-seq data were manually curated. Following this, meta-analysis was done. In conjunction with meta-analyzed data from ChIP-Atlas, data were visualized and evaluated.

**Figure 2 biomedicines-08-00010-f002:**
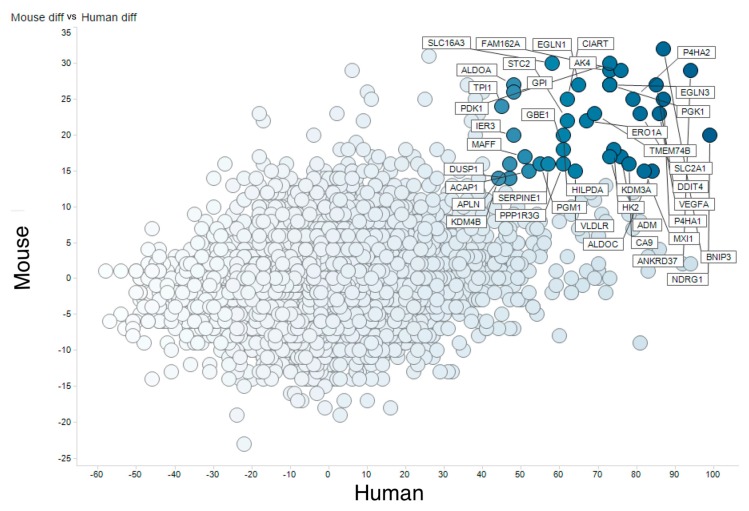
Meta-analysis of hypoxic transcriptomes by RNA-seq. Comparison of human and mouse hypoxic transcriptomes analyzed by RNA-seq. The *X* axis shows (count of human RNA-seq UP) − (count of human RNA-seq DOWN) and the *Y* axis shows (count of mouse RNA-seq UP) − (count of mouse RNA-seq DOWN).

**Figure 3 biomedicines-08-00010-f003:**
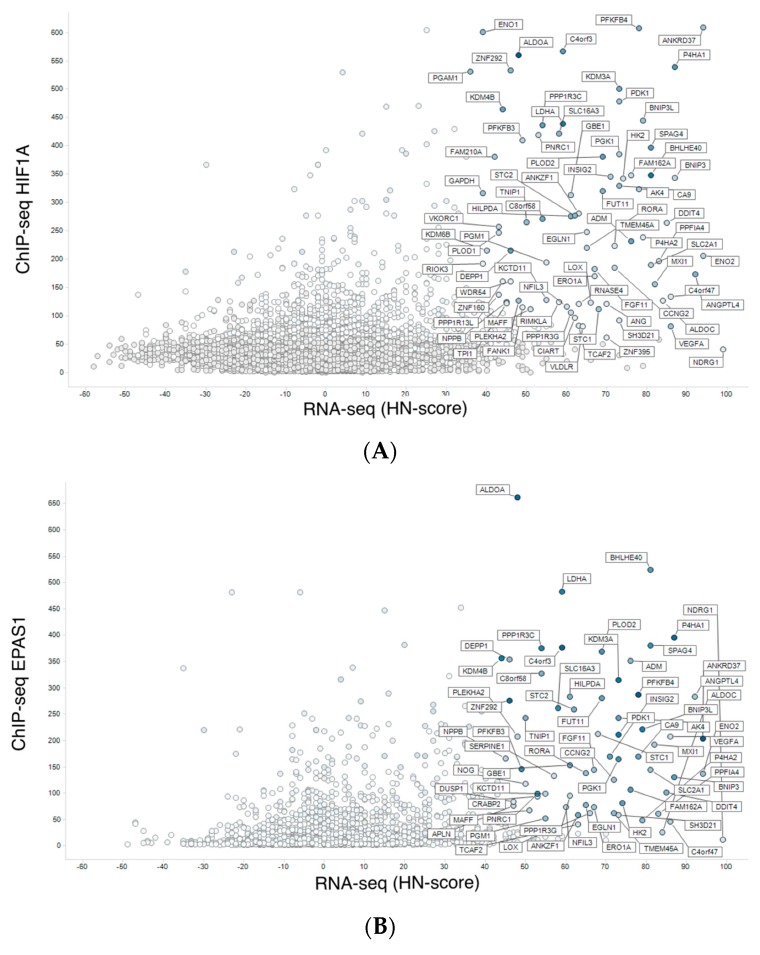
Integration of meta-analyzed ChIP-seq peak values to hypoxic transcriptomes. (**A**) Meta-analyzed hypoxic transcriptomes (RNA-seq) vs. average of HIF1A ChIP-seq peak values. (**B**) RNA-seq and EPAS1 ChIP-seq.

**Figure 4 biomedicines-08-00010-f004:**
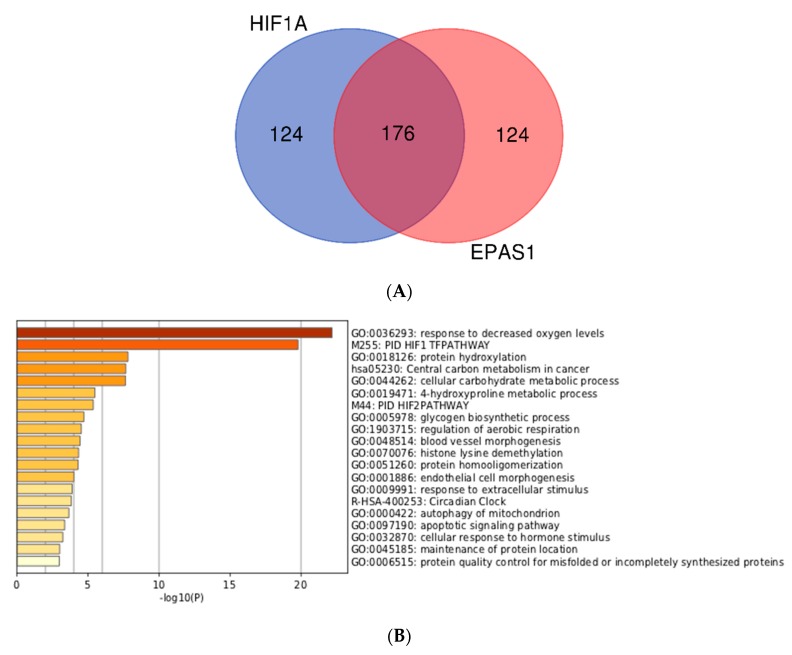
Comparison and gene set enrichment analysis of HIF1A-specific and EPAS1-specific genes. (**A**) Venn-diagram of the top 300 genes. (**B**) Enrichment analysis for 176 intersecting genes. (**C**) Enrichment analysis for 124 HIF1A-specific genes. (**D**) Enrichment analysis for 124 EPAS1-specific genes.

**Table 1 biomedicines-08-00010-t001:** List of top 25 hypoxia inducible genes. Top 25 genes with high hypoxia-and-normoxia-score (HN-score; human + mouse) with the number of paired samples that were judged as up-regulated and down-regulated after hypoxic stress. For the calculation of HN-score, see the text.

Human Gene	Human Up	Human Down	Human HN-Score	Mouse Gene	Mouse Up	Mouse Down	Mouse HN-Score	Total HN-Score
*ANKRD37*	101	7	94	*Ankrd37*	35	6	29	123
*NDRG1*	104	5	99	*Ndrg1*	22	2	20	119
*BNIP3*	92	5	87	*Bnip3*	33	1	32	119
*P4HA1*	92	5	87	*P4ha1*	27	2	25	112
*DDIT4*	94	9	85	*Ddit4*	29	2	27	112
*VEGFA*	92	6	86	*Vegfa*	23	0	23	109
*FAM162A*	82	6	76	*Fam162a*	30	1	29	105
*SLC2A1*	88	7	81	*Slc2a1*	28	5	23	104
*P4HA2*	83	4	79	*P4ha2*	27	2	25	104
*PDK1*	80	7	73	*Pdk1*	32	2	30	103
*AK4*	82	9	73	*Ak4*	31	2	29	102
*PGK1*	78	5	73	*Pgk1*	30	3	27	100
*EGLN3*	82	9	73	*Egln3*	29	2	27	100
*ALDOC*	93	9	84	*Aldoc*	17	2	15	99
*MXI1*	88	6	82	*Mxi1*	16	1	15	97
*ENO2*	99	5	94	*Eno2*	13	11	2	96
*CA9*	88	10	78	*Car9*	21	5	16	94
*ANGPTL4*	98	6	92	*Angptl4*	8	6	2	94
*ADM*	83	7	76	*Adm*	20	3	17	93
*TMEM74B*	76	7	69	*Tmem74b*	24	1	23	92
*HK2*	82	8	74	*Hk2*	21	3	18	92
*EGLN1*	69	4	65	*Egln1*	27	0	27	92
*BNIP3L*	84	5	79	*Bnip3l*	12	0	12	91
*KDM3A*	78	5	73	*Kdm3a*	18	1	17	90
*C4orf47*	91	5	86	*1700029J07Rik*	7	3	4	90

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
