# Peer review of "Meta-Analysis of Hypoxic Transcriptomes from Public Databases"

_biomedicines, 2020, doi:10.3390/biomedicines8010010_

Round 1

Reviewer 1 Report

The work by Hidemasa Bono and Kiichi Hirota makes use of publicly available hypoxic transcriptomes and genome-wide binding profiles of Hypoxia Inducible Factors to identify common

hypoxia-responsive genes and HIF-direct targets. The work is potentially interesting as it integrates the results from a large number of experiments to produce a universal picture of the transcriptional response to hypoxia. However, several important issues regarding the methodology used to analyze the data question the validity of some of the conclusions of this work.

Major comments:

1. Pag 3 lines 112-113, sentence “For the up-/downregulated gene classification, several thresholds were tested to optimize the calibration.” What tests were used for optimization/calibration? What criteria was used to select the 2-fold threshold?

In addition, the classification of genes into groups (up-, down-regulated and unchanged) based on the HN-ratio and threshold lacks statistical rigor and departs from the established workflows for differential gene expression analysis. More specifically, there are several well-established tools to determine statistically significant differential expression (e.g. Cuffdiff, DESeq). What is the reason to not to use them?

Since the HN-ratio together with the threshold is the basis for the meta-analysis, the authors should provide a detailed justification for their approach or re-do their analysis using standard methods for the analysis of differential gene expression.

2. In the analysis of the ChIP-seq data the authors chose a 10Kb window centered at the TSS. While I understand the need to set a window in order to assign ChIP-peaks to genes, this strategy is only valid for TF that tend to bind to promoter regions (such as HIF1A). However, it should not be used for TF that preferentially bind to distant regulatory regions as is the case for EPAS1/HIF2A (See reference EMBO Rep. 2019 Jan;20(1). pii: e46401. doi: 10.15252/embr.201846401. Epub 2018 Nov 14). Authors should discuss and acknowledge this important limitation of their method. In this regard the statement “Genes with high HN-score but low ChIP-seq score were also found, and these may belong to the non-HIF1 regulation pathway” on page 6 lines 208-209 must be revised (also page 9 lines 267-271). In fact, these genes could be direct HIF-targets whose induction is mediated by distant (enhancer) sites. Similarly, the preferential binding of EPAS1 at distant sites could explain why authors did not find EPAS1-specific genes (page 6 lines 212-213).

3. Page 3 114-115, Authors indicate “The number of experiments in which genes were up/down regulated was counted for all genes in the human genome.” For those experiments with several replicates, Did they count each replica as an independent “experiment”? Otherwise how did they aggregate the replicas to compare with threshold? The sum of up+down+unchanged is 128 for each human gene (table human18_hn2.tsv), thus it seems like that each replicate was considered as an “experiment” for the sake of calculation of the “HN-score”. Please clarify all this issues and substitute the term “experiment” for another that better reflects the entity that is being counted, I suggests “paired samples”. On page 4 lines 151-153 it is clearly explained and the term “paired samples” used instead of experiment.

4. The HN-score thresholds (32 for upregulation and 34 for repression) are based on the arbitrary decision of taking the 1% most extreme genes. Perhaps authors could use standard meta-analysis methods to select/score genes from a number of independent studies.

Minor comments:

1. Pag 2 line 46-47, It is unclear the meaning of “oxygen sufficiency (hyper oxygen)” , did the authors mean hyperoxia? Re-oxygenation?

2. Authors should provide an on-line accessible table with the HN-ratio for each pair of hypoxic/Normoxic paired samples, as they did for transcript quantification (https://doi.org/10.6084/m9.figshare.9948170.v1) and HN score (https://doi.org/10.6084/m9.figshare.9958169.v1).

3. Given the imbalance between the number of human and mouse experiments, adding both scores to compute the total “HN-score” results in a bias toward human-specific genes. To avoid is species-specific scores could be normalized prior adding them.

4. Figure 3 suggest that HIF binding directly mediates gene upregulation while gene downregulation is not. This pattern has been previously described (references J Biol Chem. 2009 Jun 19;284(25):16767-75. doi: 10.1074/jbc.M901790200. Epub 2009 Apr 21. and Nucleic Acids Res. 2010 Apr;38(7):2332-45. doi: 10.1093/nar/gkp1205. Epub 2010 Jan 8.) and deserves discussion an acknowledgment of previous works.

5. The preferential regulation of glycolysis by HIF1A (page 7 lines 227-228) has been previously described. This should be acknowledged including citation of original work (Reference Mol Cell Biol. 2003 Dec;23(24):9361-74.).

6. Table 1 (cited on page 5 line 177) is not present in the version of the manuscript for review.

7. The link for the online supplemental material (www.mdpi.com/xxx/s1) is broken.

Author Response

Response to Reviewer1

> Major comments:

> 1. Pag 3 lines 112-113, sentence “For the up-/downregulated gene classification, several thresholds were tested to optimize the calibration.” What tests were used for optimization/calibration? What criteria was used to select the 2-fold threshold?

> In addition, the classification of genes into groups (up-, down-regulated and unchanged) based on the HN-ratio and threshold lacks statistical rigor and departs from the established workflows for differential gene expression analysis. More specifically, there are several well-established tools to determine statistically significant differential expression (e.g. Cuffdiff, DESeq). What is the reason to not to use them?

> Since the HN-ratio together with the threshold is the basis for the meta-analysis, the authors should provide a detailed justification for their approach or re-do their analysis using standard methods for the analysis of differential gene expression.

In the previous meta-analysis study for microarray data (https://www.biorxiv.org/content/10.1101/267310v1), we used a 1.5-fold threshold, which was tentatively determined by many data analyses on microarray data. In our current study, we analyzed transcriptome data by  RNAseq. Thus, several parameters (1.5, 2, 5, 10-fold) were tested to classify up/down-regulated genes. With a strict threshold (10-fold), many experiments were classified as unchanged, and we cannot retrieve useful information from that. On the other hand, a low threshold (1.5-fold) was used, false-positives for up/downregulated were increased. Therefore, we selected 2-fold for this time. The description about this test was added to the manuscript (Page 3 lines 140).

“ For this study, we adopted a two-fold threshold after several parameters (1.5, 2, 5, 10-fold) was tested to classify up/downregulated genes.”

We acknowledge several programs for ‘regular’ DEG analysis, including DESeq2. However, our study on meta-analysis for hypoxic transcriptomes aimed to extract hypoxia-inducible genes from heterogeneous RNA-seq data (single-end and paired-end) archived in huge and heterogeneous public databases  (NCBI Gene Expression Omnibus and EBI ArrayExpress) from various laboratories by counting upregulated and downregulated genes. They are thus derived from different populations, analysis platforms, and sampling conditions, and the resultant variability can compromise meaningful comparisons, as described in the Discussion (Page 10 line 431). In other words, our study is not a ‘regular’ DEG analysis to extract statistically significant DEGs. This issue was added to the manuscript (Page 10 lines 449).

“we determined a threshold of two-fold for this study to filter genes for up/downregulated although it is not a process to extract statistically significant differentially expressed genes.”

We used Salmon (Patro, R.; Duggal, G.; Love, M.I.; Irizarry, R.A.; Kingsford, C. Salmon provides fast and bias-aware quantification of transcript expression. Nat Methods. 2017, 14, 417-419.) for quantification of transcripts, and we believe that it is a widely used method for RNAseq analysis currently. The reference for this paper was added as [11].

> 2. In the analysis of the ChIP-seq data the authors chose a 10Kb window centered at the TSS. While I understand the need to set a window in order to assign ChIP-peaks to genes, this strategy is only valid for TF that tend to bind to promoter regions (such as HIF1A). However, it should not be used for TF that preferentially bind to distant regulatory regions as is the case for EPAS1/HIF2A (See reference EMBO Rep. 2019 Jan;20(1). pii: e46401. doi: 10.15252/embr.201846401. Epub 2018 Nov 14). Authors should discuss and acknowledge this important limitation of their method. In this regard the statement “Genes with high HN-score but low ChIP-seq score were also found, and these may belong to the non-HIF1 regulation pathway” on page 6 lines 208-209 must be revised (also page 9 lines 267-271). In fact, these genes could be direct HIF-targets whose induction is mediated by distant (enhancer) sites. Similarly, the preferential binding of EPAS1 at distant sites could explain why authors did not find EPAS1-specific genes (page 6 lines 212-213).

Thank you very much for your comments. We agree with the suggestion that the binding sites of EPAS1/HIF2A are very distant from the transcription start sites from data in Figure 6 in EMBO Rep. 2019 Jan;20(1). pii: e46401. doi: 10.15252/embr.201846401. But, the description “Genes with high HN-score but low ChIP-seq score were also found, and these may belong to the non-HIF1 regulation pathway” on page 6 lines 208-209 was intended for HIF1, it is not the case. So, we added a few additional description for that. Page 7 line 394, “In Figure 3A, genes with high HN-score but low ChIP-seq score were also found, and these may belong to the non-HIF1 regulation pathway.”

We acknowledged the limitation reviewer pointed out and added the descriptions about that in the Discussion. Page 11 line 471, “By differentiating genes in this way, we can generate additional hypotheses about pathways regulating differential regulation until the hypothesis that the transcription factor binds near to the transcription start site (TSS) is valid. For example, EPAS1/HIF2A preferentially binds to distant regulatory regions [22]. In this case, we cannot use meta-analyzed ChIP-seq data for further analyses. In other cases where transcription factor binds to genomic region near TSS, ..”

> 3. Page 3 114-115, Authors indicate “The number of experiments in which genes were up/down regulated was counted for all genes in the human genome.” For those experiments with several replicates, Did they count each replica as an independent “experiment”? Otherwise how did they aggregate the replicas to compare with threshold? The sum of up+down+unchanged is 128 for each human gene (table human18_hn2.tsv), thus it seems like that each replicate was considered as an “experiment” for the sake of calculation of the “HN-score”. Please clarify all this issues and substitute the term “experiment” for another that better reflects the entity that is being counted, I suggests “paired samples”. On page 4 lines 151-153 it is clearly explained and the term “paired samples” used instead of experiment.

In this study, those experiments with replicates were counted as an independent experiment.

The complete list of used data in this study is available as described in page 2 lines 102-104 “The complete lists of these paired data are available from figshare (human https://doi.org/10.6084/m9.figshare.5811987.v2 ; mouse https://doi.org/10.6084/m9.figshare.9948158.v1)."

For the clarification as you pointed out, we changed the word ‘experiment’ to ‘paired samples’. Thank you very much for your suggestion.

> 4. The HN-score thresholds (32 for upregulation and 34 for repression) are based on the arbitrary decision of taking the 1% most extreme genes. Perhaps authors could use standard meta-analysis methods to select/score genes from a number of independent studies.

The HN-score thresholds (32 for upregulation and 34 for repression) described here were only used for enrichment analysis as shown in Figure S1. There is no standard selection method to extract genes for enrichment analyses, and we believe that the selection of 1% most extreme genes can be a standard way for such extraction.

> Minor comments: 

>1. Pag 2 line 46-47, It is unclear the meaning of “oxygen sufficiency (hyper oxygen)” , did the authors mean hyperoxia? Re-oxygenation?

Thanks for your suggestion. We meant hyperoxia, and convert ‘hyper oxygen’ to ‘hyperoxia’ (Page 2 line 56).

> 2. Authors should provide an on-line accessible table with the HN-ratio for each pair of hypoxic/Normoxic paired samples, as they did for transcript quantification (https://doi.org/10.6084/m9.figshare.9948170.v1) and HN score (https://doi.org/10.6084/m9.figshare.9958169.v1).

They are described in the original manuscript, but the description about that was unclear. Thus, we rewrite the manuscript. Page3 line 148 “Full lists of counts (up/down/unchanged) with HN-scores for all genes were accessible from figshare (human: https://doi.org/10.6084/m9.figshare.5812710.v3 ; mouse: https://doi.org/10.6084/m9.figshare.9948233.v2).".

> 3. Given the imbalance between the number of human and mouse experiments, adding both scores to compute the total “HN-score” results in a bias toward human-specific genes. To avoid is species-specific scores could be normalized prior adding them.

We acknowledge the bias reviewer pointed out. Thus, we visualized the data in the scatter plot for human vs mouse as shown in Figure 2. The numeral addition of these two HN-scores (human and mouse) was for listing as a table (Table1), and not used in further analyses. It was not intended to rank genes, but to list up those for the table. The point is that the list enriched with well-known hypoxia-inducible genes.

> 4. Figure 3 suggest that HIF binding directly mediates gene upregulation while gene downregulation is not. This pattern has been previously described (references J Biol Chem. 2009 Jun 19;284(25):16767-75. doi: 10.1074/jbc.M901790200. Epub 2009 Apr 21. and Nucleic Acids Res. 2010 Apr;38(7):2332-45. doi: 10.1093/nar/gkp1205. Epub 2010 Jan 8.) and deserves discussion an acknowledgment of previous works.

Thanks for your suggestion. In the Discussion, we added the description.about HIF biding preference by citing two papers reviewer indicated. Page 11 line 467, “HIF binding directly mediates gene upregulation, but not for gene downregulation as described in previous works [20,21].”

> 5. The preferential regulation of glycolysis by HIF1A (page 7 lines 227-228) has been previously described. This should be acknowledged including citation of original work (Reference Mol Cell Biol. 2003 Dec;23(24):9361-74.).

Thanks for your suggestion. We added a description about this and citation for that in corresponding section. Page 9 line 421, “This preferential regulation of glycolysis by HIF1A) was previously described from microarray data [19]”

> 6. Table 1 (cited on page 5 line 177) is not present in the version of the manuscript for review.

We are very sorry that Table 1 was not included in formatted version of the manuscript as it was in a separate file. We correctly added Table 1 to revised version of the manuscript.

> 7. The link for the online supplemental material (www.mdpi.com/xxx/s1) is broken.

We are very sorry that that line was in the template and was not deleted. Supplemental material is Figure S1 and it was located below.

Reviewer 2 Report

The authors developed a comparative data analysis method to identify expression fluctuations and transcription factor binding regions associated with hypoxic stress. Using this method, they performed a meta-analysis of available RNA-seq data in the public DBs and gene functional analyses using HIF1 and HIF2 ChIP-seq data.

The manuscript is well written. The study provides collective analysis of gene expression and transcription factor binding information obtained from multiple studies and can prove a useful resource for the hypoxia field of research.

Comments

The manuscripts contains an analysis of the genes regulated in common or differentially by HIF-1alpha and HIF-2alpha/EPAS1. It would be useful to add in the Introduction, information regarding the HIF-1alpha and HIF-2alpha/EPAS1 subunits (differences, specific and common target genes etc.). Reference 3 reviews the role of HIFs in the crosstalk between iron homeostasis and oxygen metabolism. The authors should also cite a more general review on the cell response to hypoxia and the role of HIFs.

Author Response

Response to Reviewer2

> The manuscripts contains an analysis of the genes regulated in common or differentially by HIF-1alpha and HIF-2alpha/EPAS1. It would be useful to add in the Introduction, information regarding the HIF-1alpha and HIF-2alpha/EPAS1 subunits (differences, specific and common target genes etc.). Reference 3 reviews the role of HIFs in the crosstalk between iron homeostasis and oxygen metabolism. The authors should also cite a more general review on the cell response to hypoxia and the role of HIFs.

Thank you very much for your suggestion. We added a more general review on the cell response to hypoxia and the role of HIF below.

Keith, B., Johnson, R. & Simon, M. HIF1α and HIF2α: sibling rivalry in hypoxic tumour growth and progression. Nat Rev Cancer 12, 9–22 (2012) doi:10.1038/nrc3183

Including this reference, we added following description to the Introduction (Page 2 line 62).

“The differential regulation of gene expression by HIF-1 and HIF-2 has been studied. Both HIF-1 and HIF-2 control the gene expression of glucose transporter 1 (GLUT1) and vascular endothelial growth factor A (VEGFA). Gene expression of glycolytic enzymes such as hexokinases (HK1, HK2), phosphofructokinase (PFK), fructose-bisphosphate aldolase A (ALDOA), phosphoglycerate kinase 1 (PGK1), and lactate dehydrogenase (LDHA) controlled by mainly HIF-1. On the other hand, gene expression of erythropoietin (EPO) and POU5F1 (OCT4) is HIF-2-dependent [4].”

Reviewer 3 Report

This manuscript is interesting.

 How level is hypoxia or hypoxia condition?

 Was same tissue in human and mice compared?

The data of different tissues may become to be different pattern.

Author Response

Response to Reviewer3

> This manuscript is interesting.

> How level is hypoxia or hypoxia condition?

> Was same tissue in human and mice compared?

> The data of different tissues may become to be different pattern.

Thank you very much for your comment.

Most used percentage of the level of hypoxia was 1%. For samples, various types of  cell lines were used. Full list of used samples is listed as online supplemental table at https://doi.org/10.6084/m9.figshare.5811987.v2 for human; https://doi.org/10.6084/m9.figshare.9948158.v1 for mouse.